# A Phenomenological Account of HIV Disclosure Experiences of Children and Adolescents from Northern and Southern Ghana

**DOI:** 10.3390/ijerph16040595

**Published:** 2019-02-18

**Authors:** Seth Christopher Yaw Appiah, Inge Kroidl, Michael Hoelscher, Olena Ivanova, Jonathan Mensah Dapaah

**Affiliations:** 1Center for International Health, Ludwig-Maximilians University, 81377 Munich, Germany; 2Division of Infectious Diseases and Tropical Medicine, University Hospital, Ludwig-Maximilians-Universität München, 81377 Munich, Germany; ikroidl@lrz.uni-muenchen.de (I.K.); hoelscher@lrz.uni-muenchen.de (M.H.); olena.ivanova@lrz.uni-muenchen.de (O.I.); 3Department of Sociology and Social Work, Faculty of Humanities and Social Sciences, Kwame Nkrumah University of Science and Technology, 00000 Kumasi, Ghana; jmdapaah@gmail.com

**Keywords:** HIV, AIDS, disclosure, children, adolescence, phenomenology, experiences, Africa, Ghana

## Abstract

Disclosure of HIV status to infected children, though challenged by caregiver dilemma, remains central in achieving the United Nations Programme on HIV and AIDS (UNAIDS) global goal of 90/90/90. This study explores children’s HIV disclosure experiences across Northern and Southern Ghana. A qualitative interpretative phenomenological design facilitated the recruitment of 30 HIV positive disclosed children and adolescents aged 9–19 years in 12 antiretroviral treatment (ART) centers in Northern and Southern Ghana between January 2017 and June 2018. Data was collected via in-depth interviews. We used phenomenological analysis applying concepts and categories identification, patterns and interconnections searching, mapping, theme building and constant comparative technique to draw conclusions. Disclosure of HIV status to children occurred with little or no preparation. Caregivers intentionally or out of dilemma often prolonged or postponed disclosure to when children aged older. Illness severity and disease progression principally defined the need for disclosure. Children preference for early status disclosure averaged at age 10 was demonstrated despite the initial disclosure experience of shock and disappointment. There was improved medication adherence despite the challenge of limited knowledge about HIV transmission, financial difficulty and food insecurity. Context and culturally adapted pre- and post- disclosure guideline laced with social protection package is needed to support HIV positive children.

## 1. Introduction

The HIV/AIDS epidemic has spread across continents, during the last three decades, with about 36.9 million people living with the infection out of which 1.8 million are children less than 15 years old [1]. Sub-Saharan Africa contributes to about 70% of the global HIV infection burden [2]. About 74% of all 1.5 million AIDS-related deaths in 2013 were recorded in Sub-Saharan Africa, despite the substantial progress in access to antiretroviral therapy (ART) [2]. In 2017, it was estimated that 21.7 million were receiving ART globally, representing 59% of the 36.9 million people living with HIV and AIDS [3]. 

Out of the global number of persons receiving ART, only 43% of HIV infected children are on ART due to many factors, but partly also because their status has been kept secret by caregivers denying children’s access to ART [3]. Despite the low HIV prevalence in most West African countries, there has been recent increase in new infections in Ghana. Though the prevalence in Ghana was stable with 1.7% in 2008, 1.3% by 2013 and 1.4% in 2014; the recent past has witnessed increase in new infections with prevalence rising to 2.4 % in 2016, according to the 2016 HIV sentinel report [4].

There were over 310,000 Ghanaians infected with HIV in 2017, out of which 28,000 were children aged 0 to 14 years [4]. ART coverage in Ghana remains very low at 34%, especially in children, though ART coverage for children had increased from 22% since 2014 [5,6] to 26% (8545 children) in 2015 and is estimated to reach 85% of all children and adolescents according to the recent data from National Paediatric Acceleration Plan for HIV 2016–2020 [5]. 

An estimated 350,000 HIV children were infected through mother-to-child HIV transmissions at birth (MTCT) across many low- and middle-income countries [6] before Prevention of Mother to Child Transmission of HIV (PMTCT) services were introduced. A scale up in PMTCT activities has been implemented in Ghana since 2009.

In 2008, Ghana in particular witnessed a high prevalence of 2.9 % among women attending antenatal clinics with 3700 children newly infected with HIV compared to the national prevalence of 1.7 % that same year [6]. PMTCT has led to a decline in the number of children born who are HIV infected. Additionally, early ART initiation has led to the improved survival of HIV-infected children, notwithstanding new challenges have emerged as children grow into adolescence.

Many children grow into adolescents without knowledge of their HIV infection. This situation endangers many of these adolescents and their potential partners as it promotes physiological distress among those who might have heard of the danger part of the infection in school [7]. Increased risk behaviour such as medication non-adherence, substance use and sexual risk taking have been associated with adolescent HIV infection for status naïve children [7,8]. Secrecy surrounding HIV treatment coupled with limited control over the living environment are significant barriers to adherence.

Disclosure to children remains critical for addressing drug adherence and pre-knowledge of HIV transmission risk for adolescent on first sexual initiation. The goal of paediatric disclosure is for children to know their own HIV status [9]. Many caregivers fear disclosing the status of positive children to the children because of the anticipated distress that may come along with it for both the child and the caregiver. Child HIV status disclosure is complex and brings about hesitancy and ethical dilemmas taking into consideration the socio-cultural and stigma related issues and secrecy surrounding HIV infection [10]. As a result, caregivers have a dilemma whether to disclose or not to to a child the child’s own HIV/AIDS status, principally due to their inability to trade-off between the benefit and challenges that comes with paediatric disclosure considered to contribute to increasing child survival [11,12]. Disclosure offers a psychological boost, facilitates better coping strategies for the child and gives a protection mechanism for potentially early sexual initiation and risky sexual behaviour [11,13].

According to the American Academy for Paediatrics, disclosure of child HIV status ought to be individualized. This should consider all issues surrounding the child’s cognitive ability, developmental stage, HIV clinical status and social circumstances after adequate counselling of parents and caregivers of HIV-infected children by a health professional about disclosure to the child that their infection has occurred [14]. 

According to WHO, children of school age should be told of their HIV status; and this should be done incrementally. This is to accommodate the age and child development specific needs of the child. Children cognitive skills and emotional maturity have implications for a full disclosure [15]. Age of child, perceived cause of HIV, child’s inability to keep diagnosis to self, the stigma attached to HIV and fear of physiological harm has been noted to prevent caregivers and, in some instances, health care providers from disclosing HIV status to infected children and adolescents [16,17].

Theoretically, the phenomenological approach anchors the study. Phenomenology is considered as a reflective analysis of life-world experiences and situations [18]. The justification for situating this study within this theoretical context is because, the perspective helps chart an understanding of and meanings to human experiences [19] and/or explore concepts from new and fresh perspectives [20,21]. The nature of this study is better be explained by the phenomenological theoretical position. It is anticipated that it will allow the researchers gain insights into social phenomenon of living with HIV and AIDS post disclosure and reveal the “essence of things” in real world. 

In Ghana, guidelines on child/adolescent chronic disease disclosure remains sketchy and almost non-available despite the continued physiological distress that confronts children and adolescent living with HIV. There is inadequate information about experience and challenges adolescent encounter prior and post disclosure. This study explores the lived account and narratives of HIV disclosed children and adolescents from diverse backgrounds, cultures and context and geographically distinct locations in the Northern and Southern Ghana. The findings of this study contributes both to theory and practice of post disclosure experience of children moving beyond caregiver account to lived experiential account by infected children. 

## 2. Materials and Methods 

### 2.1. Study Setting

The study was conducted in three regions in Ghana. These are the Ashanti, Northern and Upper East regions of the country. By the operational definition of this study, the Ashanti region is considered as the southern part of Ghana whilst the two remaining regions are classified as Northern Ghana. In 2017, the Ghana AIDS Commission reported the regional HIV prevalence, of which the Ashanti region had a share of 3.2%. The HIV prevalence rate recorded for the Upper East region was 1.7%, with the Northern region recording the lowest national prevalence of 0.7% [4]. The regions and the study areas are indicated in Figure 1. The study areas are categorized under each of the regions as both urban and rural.

The sites were the HIV clinics of the Bongo and Tongo district hospitals as rural sites and the War Memorial Municipal Hospital in Navrongo and the Bolgatanga Regional Hospital as urban sites in the Upper East region. In the Northern region, the selected urban sites were the Tamale Teaching Hospital (TTH) and the Tamale Central Hospital. The Walewale district and the Savelugu district hospitals were selected as the rural sites for the Northern region. In the Ashanti region, the urban hospitals selected were the ART units of the Suntreso and Tafo government hospitals whilst the rural sites chosen were the ART units of the St Patrick Hospital (Offinso) and the Presbyterian Hospital at Asante Akyem Agogo. 

### 2.2. Study Design 

The study utilized entirely qualitative methods employing in-depth interviews (IDI) to have a comprehensive understanding of the lived experiences of HIV positive disclosed children. Qualitative study primarily focuses on the meanings and hinges on the conceptual level rather than on numbers or statistics [22,23]. Thus the design’s focus is more meaning oriented. Using phenomenology [21] as a qualitative research approach, the focus of the study was to understand the meanings and the experiences positive HIV status disclosed children from their own perspective. This ranged from individual to situational experiences through to their coping strategies and personal transformation after their status disclosure. The notion of transition from a state of unknown status to a state of known status constitute, what Prescotte and Hellstén [24] explain as an advancement from the known or familiar to the unknown and the acceptance of new cultural, social, and mental challenges. This reflects the situation of HIV positive children in the study setting.

### 2.3. Sampling and Participant Recruitment 

Recruitment of study participants was carried out by means of purposive sampling methods facilitated by the first author with the assistance of the trained ART unit heads. In total, 30 children and adolescents aged 9–19 were recruited from 12 ART sites from northern and southern Ghana. The 12 ART sites were selected to represent the diversity in the HIV prevalence, understanding and context. The rural and urban sites were representative of the size of the ART units and variation in the number of clients and distance from study site to the regional capital. Pre-meeting arrangements were made by the principal researcher and the facility managers with children who met the eligibility criteria, thus children with known status and caregiver consent in a convenient location.

### 2.4. Data Collection 

Data collection spanned from January 2017 to June 2018. Due to the difficulty in reaching out to the children mostly in their rural communities, participants were often engaged during the clinic visit days when the children came to the ART clinic for their drugs and for general check-up. In centres where drugs were given in advance of three months to accommodate for the financial difficulty of caregivers coming to the facility on weekly or fortnightly, the principal researcher had to wait for three months intervals to be able to reach the children on their clinic days.

A trained focal person supported the conduct of the interviews using structured interview guides. The focal person in each clinic additionally served as a translator for children who could not understand English. The language used was principally English, followed by Asante Twi, with other diverse languages which included Grune, Mampruli and Dagaare. In the northern parts of Ghana (Northern region and Upper East region), the principal investigator spent 10 days on average in each of the eight ART centres. Field notes were taken by the principal investigator with the support of a trained research assistant. 

In the Southern part, the materials were translated into the Asante Twi language. In the Northern regions, the translations were made verbally into the local dialects that were unique to each ART site. The interviews were conducted using direct discussions, focusing on local explanations for the causes of HIV, explanations given to the cause of the illness condition prior to disclosure, process of disclosure, post disclosure experience, ART adherence and social support. Stigmatization encounter, community and household level attitudes towards the children were also explored.

### 2.5. Data Processing, Coding and Analysis

The phenomenological analysis characterized by its intuition and reflective nature and anchored on reading textual narratives intensively and repetitively [20,24]. The interviews were audio-recorded and supported with field notes and later transcribed. Two trained qualitative researchers were engaged to perform an independent transcription which were compared and corrected to guarantee the reliability of the data. This was followed by introspection and an eidetic reduction process [24]. The transcription followed a constructivist approach through the viewing of reality and experience from the perspectives of the children. 

This approach according to Creswell and Miller [25] allows for the quality of collected data and accurate reporting of the findings of the study. The constructivist approach was supported with a content analysis. The content analysis followed O’Leary’s [25] six steps or procedure for analyzing qualitative data. This included: reading through data; organizing and coding; searching for patterns and interconnections; mapping and building themes; building thematic data; and, drawing conclusions. The raw data was read more than once and in some cases several times by two researchers initially. Agreed upon codes were used to delineate common observations. The common patterns were identified independently by the two researchers. The emerging themes were then searched for by each researcher after which they were all grouped. In building the thematic data, four of the researchers were engaged to examine the theme data built. The conclusion drawing involved a constant revisit to the field notes, identified and matched patterns, replay of the audio recordings and constant feedback from the researchers. The researchers developed the coding scheme to help maximize the breadth and depth of the analysis.

After transcription, there was the adoption of open coding technique. The open coding procedure involved concept identification and categories segmentation of data to smaller units after which they were labelled, and their conceptual properties described. Though this can be done word-by-word, line-by-line, by paragraphs, or by perusing of the entire transcripts, we adopted an eclectic approach by combining different approaches [21].

Emerging new subthemes which were constantly compared with the data. Any inconsistencies in coding identified were resolved by going back to play the audio, re-read the transcripts and checking of records from the field notes. Identical quotes supported the major or sub-themes on the basis of similarities and contrasts. The new themes that emerged were presented as the study findings through interpretative reasoning and narration, whilst constantly linking codes to the sub-themes. The posteriori inductive approach primarily guided the analysis by making inference on the implications the findings have for the phenomenological theory [26,27,28].

### 2.6. Quality Control and Data Rigor 

The data were collected by trained social scientist/medical sociologist using qualitative phenomenological data collection techniques with the help of language translators in clinics where study participants could not either speak or understand English or Asante Twi (dominant language spoken in Ghana). Field supervisors who supported the data collection were trained persons working within the ART clinics. An additional person recorded the interview, whilst field note-taking continued. 

### 2.7. Ethical Statement 

The ethical approval for the study was granted by the Ghana Health Service Ethics Review Committee with ethics approval number GHS-ERC: 05/06/17 and the Ethical Committee of LMU Munich (Project Number 18-018). Parent/caregiver consent was obtained for all study participants. Child assent was also sought. This was led by the principal researcher and the research assistants who sought for study participants willingness and explored how they preferred to be interviewed using the information sheet and consent/assent form. 

## 3. Results

### 3.1. Participant’s Background Information, Current Preoccupation and Living Arrangement

Thirty HIV positive disclosed children aged between 9 and 19 were enrolled in the study: 17 females (57%) and 13 (43%) males. The average age of the children and adolescents was 14 years. Many children [13] mainly lived with their mothers. Four of the children stayed exclusively with their grandparents and the remaining lived with their siblings and distant relatives. Five children did not have their biological mothers, whilst eight had both parents’ dead, with three having their father demised. 

Across the geographical belts, thus both northern, southern, rural and urban, there was no child who reported of currently staying with a father. Table 1 presents an overview of the characteristics of study participants in terms of site or recruitment, urban rural dwelling and the region where child lives. 

All the children were preoccupied mainly with two domains. Either children or adolescents were currently in school (25/30) or were engaged in apprenticeship or offering support to their parents/caregivers owning shops and small businesses. One adolescent was enrolled in tertiary education, two in secondary schools with the remaining currently enrolled in basic education (junior high school and primary school).

Field and participant observations of the children and adolescents show most of the children to be physically lean and looking impoverished. Generally, among those in school, majority of them were far late in their educational grade. When we asked the children about their classes matching it against their age (average age of beginning basic school being 6 years in Ghana), the following responses were offered:
“…I am 14 years … I live with my mother and siblings …I am in school and in class 4”.(Female, 14 years, Agogo, Southern Ghana)
“I am 16 years; from [ ] …I am in school; and in class 4; will be progressing to class 5”.(Female, 16 years, Agogo, Southern Ghana)

Similar observations were made among participants who were in the junior high school level:
“…I am 17 years … I am in school, in junior High School Year 3”.(Female, 17 years, Tafo, Southern Ghana)

Few of the children were in the class which corresponded to their age. 

### 3.2. Study Emerging Themes 

Central to the themes that emerged were: (i) Pre-disclosure knowledge on HIV and AIDS, (ii) the disclosure approach and process; (iii) earliness or lateness of disclosure nested in preferred ideal age of disclosure; (iv) reaction and response to disclosure news; (vi) self- disclosure of status to peers; (vii) medication intake and adherence and (viii) coping with HIV and AIDS. The themes are displayed in Table 2.

### 3.3. Pre-Disclosure Knowledge on HIV and AIDS 

Despite the relevance of knowledge on HIV and AIDs, many participants demonstrated limited understanding on HIV and AIDS with circumstances forcibly compelling their parents and guardians to disclose HIV status to the young ones at a later stage.
“My grandmother informed me after I persistently ask her why I have been taken [this] medicine every time and going to hospital almost every week. At first, she told me that my disease is scary and that if I don’t take the medicine, there will be rashes all over my body and I will die prematurely. Then I ask her the name of the disease. She calls me privately in her room and then disclose to me”.(Female, 14 years, Agogo, Southern Ghana)

There was however, a disparity in in low and high knowledge of HIV and disclosure related information between participants from southern and the northern parts of the country respectively. The observation is illustrated in Table 3.

The sub-theme of how children contracted HIV emerged prominently. Two dominant narratives that featured in their attempts to provide response to how they became HIV positive were either “*I do not know*” or an incidental explanation that concerned the response given to them by their caregivers in their attempt to find answers.
“…*I don’t know*”.(Child 1, Child 3, Child 4, Child 7, Child 8, and Child 14)

Only one participant knew about how she got infected with HIV:
“…*actually from my step mum* [was narrated to child by a step mother], *they say it was through my mum and my dad*”.(Female, 18 years, Tamale, Northern Ghana)

Most of the children indicated circumstances around their illness as the time around which they got to know about their HIV infection. Giving incidental account of participant’s recollection of their first conscious encounter with HIV and AIDS related illness some shared.
*“…During the time I was in school, I felt sick and was rushed to the hospital. Then the doctor discovered it, I was about 12 years by that time”*.(Female, 18 years, Tamale, Northern Ghana)
*“I can’t remember. But I had a boil and came to hospital. It was then that it was discovered by the doctor”*.(Male, 13 years, Tamale, Northern Ghana)

Some of the children were knowledgeable on the potential source of HIV infection despite being unable to isolate which among the sources could be linked with their own infection. When asked, few of the participants were accurate with their responses:
*“…by having unprotected sex and secondary taking used blade, and transfer from parent to children”*.(Female, 16 years, Tafo, Southern Ghana)

### 3.4. Disclosure Approach and Process

Disclosure of HIV status to the participants was often postponed to later years, deferred, lied to about, replaced with excuses or partially performed and at best without any child preparation. In most instances, disclosure became possible when the illness aggravated and there was the urgent need to tell the child. With the exception of few participants whose disclosure had taken place at very young age specific to their development, majority of the children had unprepared disclosure and felt very disappointed. The participants explained how their status was disclosed to them:
*“The health official disclosed to my grandmother. My grandmother instructed me not to take oily foods, rice and corn related foods. I asked her why… Then she replied that it is because of my sickness. I asked what sickness?… But she refused to tell me. She then told me that if I should use razor blade and, as a result, there is a cut on me. Then if someone else uses the blade and gets a cut, the person will also be affected with my disease. So, I started thinking about it since I was taught by my science teacher that the use of infected blades or needles could cause HIV/AIDS. …Then I realized I am HIV positive, but my grandmother felt adamant and uncomfortable to disclose to me”*.(Female, 17 years, Tafo, Southern Ghana)

There were few instances where children appeared to be prepared for the disclosure. In such few instances, disclosure occurred in private between caregivers and children. Some children expressed satisfaction about the approach and privacy of the disclosure amidst minor reservations about the timing (lag between diagnosis and disclosure), while others remained less worried with the time (age) of disclosure:
*“My mother disclosed to me privately, in the room. I was informed too late. I think I should have been informed earlier than the time I was actually informed”*.(Female, 14 years, Bongo, Northern Ghana)
*“When my father died, my mother came to take me away from my father’s home, my mother sent me to do the lab at the hospital and I was told… about 8 years ago* [disclosure took place when the child was 7] *…It was the doctor who told me at the hospital. I did not react in any way because I was young”*.(Male, 15 years, Tafo, Southern Ghana)

### 3.5. Earliness or Lateness of Disclosure Nested in Preferred Ideal Age of Disclosure

In the considered opinion of the children, status disclosure to them was late. The reported mean age of disclosure among all the 30 children was 13 years. Some of the children had grown into later years of adolescence before being told about their HIV positive status. Among some participants aged 17 and 18, they had only been told of their status a year ago prior to the conduct of the study. Many of them showed resentment and disappointment with the age at which status disclosure took place. This resonated with participants across Southern and Northern Ghana and urban and rural sites:
*“I think it should be above 3 years now… I feel I should have been informed earlier than the time I was informed”*.(Male, 17 year, Agogo, Southern Ghana)
*“When I was 17 that they informed me*”.(Male, 18 years old male, Tamale, Northern Ghana)
*“I was informed in 2012; which is 5 years ago…was too late. My mother should have informed me. I tried several times, but she failed to disclose to me”*.(Female, 16 years, Tafo, Southern Ghana)
*“I am 17 years; from Tafo Medoma. My parents are dead. I know is a dreadful disease, I can’t remember. But I had boil and came to hospital. It was then that it was discovered by the Doctor…Not long ago. Not even more than a month”*.(Female, 17 years, Tafo, Southern Ghana)

Notwithstanding, four of the children whose disclosure occurred particularly at ages 9,7,8, and 6 were comfortable with the timing of disclosure since they had their status disclosed to them at relatively younger age despite sharing varied opinion on what early age meant to each participant:
*“When I was in class 2, I was six years… It was normal, if they did not tell me in the earlier stage maybe I would not have been alive by now, my aunty knows about it, my aunt’s first daughter, they treat me well”*.(Female, 16 year, Tafo, Southern Ghana)
*“I can’t recall but it will be about 2-3 years now…I said to her its ok…Yeah, the timing was ok”*.(Female,12 years, Navrongo, Northern Ghana)

Participants were of the view that disclosure needed to be carried out privately. Though children the ideal age of disclosure were non-uniform mostly ranging between 8-12 years, participants laid significant emphasis on four principal requirements: (1) early disclosure, (2) serene environment-social and physical for disclosure, (3) parent/caregiver frankness with information on child illness, (4) holistic preparedness of child prior to disclosure.
*“…I think disclosure should be made by examining the person’s psychological state and should not be done unexpectedly like what they did to me”*.(Female, 17 years, Tafo, Southern Ghana)
*“Privately like what my mother did; but the information should come early after status is known”*.(Male, 14 years, Agogo, Southern Ghana)
*“I think disclosure should be made immediately it is known but the patient should be made aware that HIV test is to be done. When the person gives the approval, then it can be tested and disclose to him or her”*.(Female, 17 years, Agogo, Southern Ghana)
*“…I think disclosure should be made by first encouraging the person, then you tell the person about the importance of the medication. After that, you can gradually unfold the information to the person. Whilst you tell the person, you still encourage the person that all is not lost”*.(Male, 18 years, Offinso, Southern Ghana)

Figure 2 presents an overview of the pre-post disclosure context of children and adolescents studied.

### 3.6. Immediate Reaction and Response to Disclosure News

Among nearly all the participants, sadness and disappointment greeted them with the news of hearing being HIV positive. In many instances, participants further explained that, the context under which they became knowledgeable of their illness condition worsened their reaction. Illness severity appeared to be the trigger to their being told about their HIV status.
*“It pained me so I wept that very day…… I felt unhappy”*.(Female, 14 years Bongo, Northern Ghana)
*“…I felt very sorrowful. It really pained me”*.(Female, 17 years, Agogo, Southern Ghana)
*“…yes, what they did was paining me, getting old and telling me about it, …oh am okay with that”*.(Female, 18 years, Tamale, Northern Ghana)

Those who could not make any visible reaction, explained that they had been suspicious of what their illness could be:
*“So I read the label on the box and realise that it was HIV”*.(Female, 14 years, Bongo, Northern Ghana)

A common theme that emerged was the silence of the children /adolescents to inquire from their care givers/ mothers how they became infected. In what appeared quite surprising, none of the children questioned their care givers but for two relatively old adolescents (16 and 18 years) whose disclosure by health professionals appeared to them extremely unanticipated and unprepared. One reported:
*“During the time I was in school, I felt sick and was rushed to the hospital. Then the Doctor discovered it, I was about 12 years by that time …I had ‘kooko(boils)’ and came to the hospital. Upon diagnosis by the Doctor, he disclosed to me about my HIV status. He asked me whether I am aware and I said no. I have not been informed by anyone…It really pained me. My mother then told me they shouldn’t have informed me at that very time. It should have been later. Then later, my mother approach me privately and explained everything to me concerning how I became HIV positive. She says she doesn’t know, whether it was a spiritually purchased illness for me or a blade that I might have used…Not at all. The doctor unexpectedly disclosed to me. In fact, I did not believe it but later I realized it was true. The disclosure was too instantaneous”*.(Female, 18 years, Agogo, Southern Ghana)

Participants further explained that they had not self-disclose by concealing their status from their peers:
*“No, I don’t even have friends; I have somehow restricted myself from people around …and its something my mother has warned me about it…. It is sensitive situation. I have boyfriend but have never disclosed to him. My mother told me if I tell him, the relationship will break and I also don’t want to lose him. I don’t want to tell him because he will leave me”*.(Female, 18 years, Agogo, Southern Ghana)

### 3.7. Medication Intake, Adherence and Coping Strategy

HIV medication intake among participants was often scheduled daily between morning and evening dose. Nearly all children were affirmative in being adherent to the taking of their medications with few defaulting. During circumstances of missing medication resulting from forgetfulness or busy work schedule, children often came to take the medication or called for it to be brought to them wherever they were to be taken. Nearly all the children took their medication in the home with few taking it at school or workplace. Few of the children shared their experience of abdominal pains associated with the HIV medication intake. The strongest motivation for the continuous taking of the medication was the desire to be healthy:
*“I take every day---in the mornings and in the evenings …. it is my grandmother who encourages and ensures that I take the drugs well, sometimes I forget. I feel comfortable taking drugs at home since people do not say anything bad about the drug……It is the abdominal pains”*.(Female, 16 years, Tongo, Northern Ghana)
“…one in morning and evening …it’s my life, and I have to depend on it, I just realize it’s a duty There was a day I forgot that I did not take, so they had to bring it to me, when I forgot, they had to bring it to me at work …when I forget, I quickly go and take but now I take it 7:00 in the morning …when you take the drugs, like, I don’t know how to put it, the drug is strong… No…I don’t feel anything. I don’t take the drugs at the worksite …no worse experience but when I don’t take then I have some bad experience”(Female, 18 years, Agogo, Southern Ghana)

The continuous consumption of the HIV medications does not come as a pleasant experience to the children at all times. In some instances, children questioned their own action of daily taking the HIV medication. Other children contend with the bitter taste of the drug and vomit alongside a situation that has caused some to change their initially prescribed medication:
*“I take morning two…evening two. At first, I used to vomit. That was the first drugs…Yes. I used to vomit when I take that drug. But this new one, I am feeling good with it: I take every day---in the mornings and in the evenings…I know that it will strengthen my immune system and the viruses will be destroyed……Yes. Sometimes, I feel I am taking the drugs too frequently, so I pulse in some cases. Again, taking the drug is not a pleasant experience, at the work place, I take the drugs secretly so that people will not know”*.(Female 18 years, Agogo, Southern Ghana)

The taking of the medication demanded adequate food intake which was mentioned as a challenge among many amidst financial difficulties.
*“Sometimes, I am forced to skip my weekly visits to the hospital due to limited funds …Sometimes, I don’t eat in the afternoon due to financial problems. I have financial problem, the food we eat sometimes faces shortages; so I eat in the morning and in the evenings only. I eat twice a day. I wish that government will give positive HIV and AIDS children money to go to school. I have limited finance which affects my eating, transport and general living condition”*.(Female, 18 years, Tamale, Northern Ghana)

## 4. Discussion

This study explored the HIV and AIDS disclosure and post-disclosure experiences among children and adolescents from Northern and Southern Ghana using phenomenological lenses. We critically examined the disclosure experiences, reactions and response to the disclosure and identified the context appropriate for disclosing HIV and AIDS based on the child lived experiences. This study to the best of our knowledge is the first qualitative attempt at characterizing children and adolescent post HIV and AIDS disclosure experiences in Ghana. Previous studies reporting on whether disclosure of HIV status to infected child was early or late elicited the perspectives of caregivers and not children [29,30]. Reports from caregivers on the age of disclosure have ranged from 9.3 years in South Africa, 10.39 years and between 6-10 years in Ghana [29,30,31].

The mean age of disclosure as reported by the children in this study was 13 years. Notwithstanding, the children expressed dissatisfaction with the timing of disclosure and opined that the timing for disclosure was late. Some had been informed only a few months before reaching their eighteenth birthday, despite the few who got to know their status as early as six years. An earlier study in Ghana by Gyamfi et al. reports that the preferred ideal age for status disclosure to infected children according to the care givers was 15 years. This was however quantitative and an elicited opinion of caregivers. 

Our study reports of age 10 as the ideal age children and adolescents preferred to be disclosed their status with the range being 8–12 years. Certainly, differences exist in preferred mean age of disclosure (10 years vs 15 years) between caregivers and children themselves, though in practice care givers disclosed to children at age 13 [17]. This appears to be midway age ideal for caregivers though many of the children did not demonstrate acceptance of the age therefore rating the timing of their disclosure as late. Caregiver/child crush of interest and preference appears to be emerging. This calls for the creation of friendly and open avenues for parent/caregiver–child dialogue. The topic of disclosure remaining silent until time is becoming imminent and inevitable. 

Similar to the sentiments expressed by other adolescents in different African countries like South Africa, Zambia and Zimbabwe, Ghanaian children and adolescents showed initial distress, reacted shockingly when disclosure was initiated, yet the children desired to be told and offered the correct information about their illness [32,33]. The initial shocks were accommodated by the children with the passage of time and development of resilience. There is the need for the development of a structured pre- and post-disclosure care and support intervention for adolescents and children with their primary care givers to facilitate the acceptance of their new status. Further investigation may be needed to explore the unique child characteristics that shaped their resolute disposition to the news of their status disclosure [28,33,34].

Our study found that in majority of the disclosure encounter children were not prepared, and not ready as disclosure happened instantaneously. In Zambia, similar finding has been confirmed with only 6% of children and adolescents being informed by both parents of their HIV status. The possible reason for this occurrence may be due to parents/caregiver weak-preparedness and unskilled predisposition to disclose amidst the fear of blame and stigma towards the children and child self-disclosure to peers [33,34,35,36]. This often prolonged the years or time to disclosure. This heightened fear is however allayed as evidence from the Cluver et al. study has shown how medication-adherent children with early status disclosure of less than age 12 become [29]. 

Moreover, the findings in this study emerge as a respite to caregivers for heightened fear since an estimated 97% (29/30) of the children and adolescents had never self-disclosed their HIV status to their peers or any other person outside the confines of their nucleus family nor did they report of felt or community level stigma [36]. 

The difference in disclosure approach could be attributable to the non-availability of a paediatric disclosure guideline in Ghana compared to South Africa. Caregivers have had to trade off a balance between having to lie to children, non-disclosure, incremental disclosure and full disclosure. Our findings are consistent with the picturesque narrative on paediatric HIV disclosure landscape across Sub-Saharan Africa where evidence from 180 sites in six global regions from 31 countries, only a third of the sites to have existing standardized protocol for paediatric disclosure [37]. In this study, early disclosure was to be interlaced with serene social and physical environment, frankness with their illness condition and holistic preparation of the children towards receiving disclosure news.

Consistent with findings in Kenya, Zambia and South Africa, some children blamed their caregivers except that the reasons for the parental blame by children differed from the present study compared to earlier report [28]. In the current study, children blamed their parents/caregivers by reason of their inability to have disclosed their status to them earlier and not by reason of parent/caregiver transmitting the virus to the children as reported in previous study. None of the children studied reported being bold enough to ask the source of their HIV infection. From the study findings, caregivers gently and smartly refused to tell the children how they got infected with HIV or children were not interested in knowing. 

One in three of the children lived with their mothers as primary caregivers with the remaining staying exclusively with their grandparents or distant relatives. This observation is consistent with other studies in Ghana where the family structure of Ghana has contributed to relatively huge numbers of children without parental care which is estimated at 4.7% of Ghana’s population and 10.4% of the entire children and adolescent population in Ghana [38,39,40,41].

Despite the limitedness of social support for the children in the study, it served to moderate the effect of the illness on child health. This finding has received similar expression in previous cross-sectional studies in East Africa and Latin America [35,42,43] and meta-analysis studies in Latin America and Asia [44,45], where family social support had improved health outcomes for children living with HIV and AIDS. On the contrary, the study findings are inconsistent with Doku et al. from Lower Manya Krobo District of Ghana where a weakened traditional system contributes to lower social support by families to HIV positive children in earlier study [46]. 

None of the school children reported disruptions in school attendance due to financial difficulties as compared to the recent findings from Cambodia et al [42], though peer support group was absent among the Ghanaian children despite its availability among positive children elsewhere. Generally, the children had delayed in academic progression. Children from the northern part of the country had delayed more compared to those southern Ghana. This may reflect a national situation where school enrolment among children in the northern part are less compared to that in the southern part of Ghana. This finding is in tandem with the overall general educational outlook in Ghana where less than 50% of the population in the three northern regions aged 11 years and older are literate compared to a minimum of at least 69% in other regions with the southern Ghana inclusive [47].

Our findings offer an additional rationale for improving the knowledge of the children and adolescents living with HIV and AIDS. Knowledge on HIV infection and mode of transmission among the children and adolescents was limited and varied between high and low for children from Southern and northern Ghana respectively. High knowledge on HIV and AIDS have been reported by previous studies in Ghana [48], whilst studies by Enimil et al. have established limited understanding of and knowledge on HIV transmission modes among 13–22 aged adolescents in urban Ghana [49]. The limited knowledge of children on HIV and AIDS might partly explain why caregivers remained reluctant in disclosing their status to children. Similarly, it may explain why in many instances, disclosure occurred at the hospital and by health care providers [17], contrary to the evidence from Zambia, Kenya and Zimbabwe, where disclosure took place in the either at home or hospital with a preference for at home by parents [50]. This did not vary among children living in rural or urban locations, despite the difference in prevalence and variations in the odds of infection for urban (2.3%) and rural (7%).

Our study results argue that efforts towards managing the pre- and post-disclosure interface as well as the transition from childhood to adolescence—a phase of high vulnerability, will require designing adolescent-focused interventions with multiple layers of the sociological context that is need based with comprehensive social protection packages. The call for these interventions is timely and consistent with earlier study recommendations by Enimil et al. with our findings making a case for the inclusion of the phenomenological context of the children in the intervention design [49].

### Limitations 

This study is limited by the possibility of social desirability biases. It remains unclear whether children and adolescents offered explanations that reflect the truth about their prior and post disclosure experience or gave responses they considered as what interviewers wanted to hear though this was limited by asking questions that filtered the responses and elicited consistency. The eclectic phenomenological approach used in this study aimed at presenting a lived experiential account of what participants considered as the truth on disclosure narratives worth exploring. As a result, the authors did not attempt at determining whether participant’s world view and experiences presented were the ‘truth’ as the responses given by participants were considered their version of the truth on their disclosure experience. Consistent with qualitative studies, our findings are limited by being context specific for which interpretation may have to be done with some care. Though several attempts were made to ensure accurate and pure bracketing—an essential requirement essential when using a phenomenological design study—the adoption of research translators, different study sites selection with unique cultural context, may not have been able to ensure a full compliance with pure bracketing.

## 5. Conclusions

Our study, the first ever on post-disclosure experiences among children and adolescents across Southern and Northern Ghana, demonstrates HIV and AIDS disclosure in Ghana has often occurred with children being unprepared and at very late period in the child’s development. The findings of this study provide in-depth and nuanced understanding of unexplored area of child and adolescent HIV and AIDS post-disclosure experiences from the child’s own account. Furthermore, it provides that despite the emotional, remorseful and pain that are immediately associated with HIV and AIDS disclosure among children and adolescents, they preferred to be told their HIV and AIDS status at very early stage and preferably between age 8 to 12. 

The near absence of evidence of child self-stigma, non- self-status-disclosure to peers and improved medication adherence comes as a relief to parents and primary caregivers who are often in a dilemma and cite these factors as predominant reasons for delayed status disclosure to infected children. The findings of this study highlight the varying limitedness in child adolescent knowledge on their own illness and disease aetiology across southern and Northern Ghana. The findings on the near absence of social protection packages and support groups compared to the adult population [51] coupled with financial difficulties have implications for child and adolescence engaging in risky sexual behaviour and transactional sex. Effective care for post-disclosed children and adolescents will require that care givers of HIV and AIDS infected children be provided with social, financial and income generating resources coupled with programs that aim at strengthening the resilience of the children to improve their wellbeing post-disclosure. 

## Figures and Tables

**Figure 1 ijerph-16-00595-f001:**
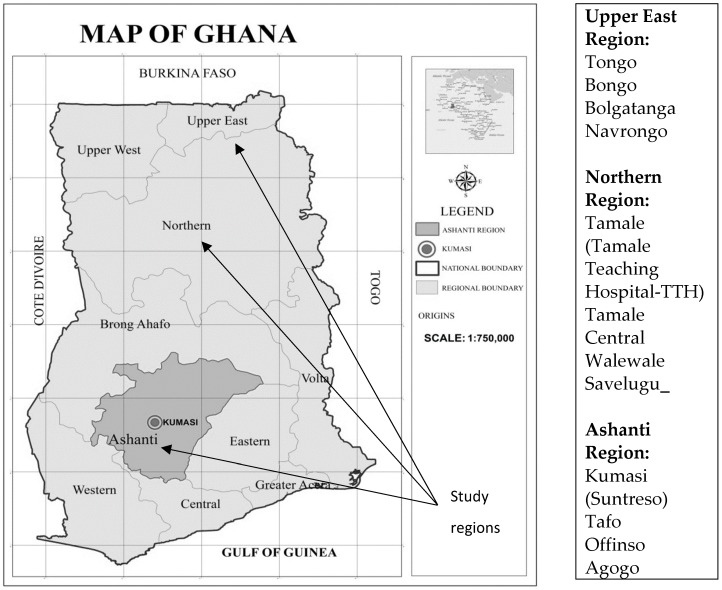
Map of the study area.

**Figure 2 ijerph-16-00595-f002:**
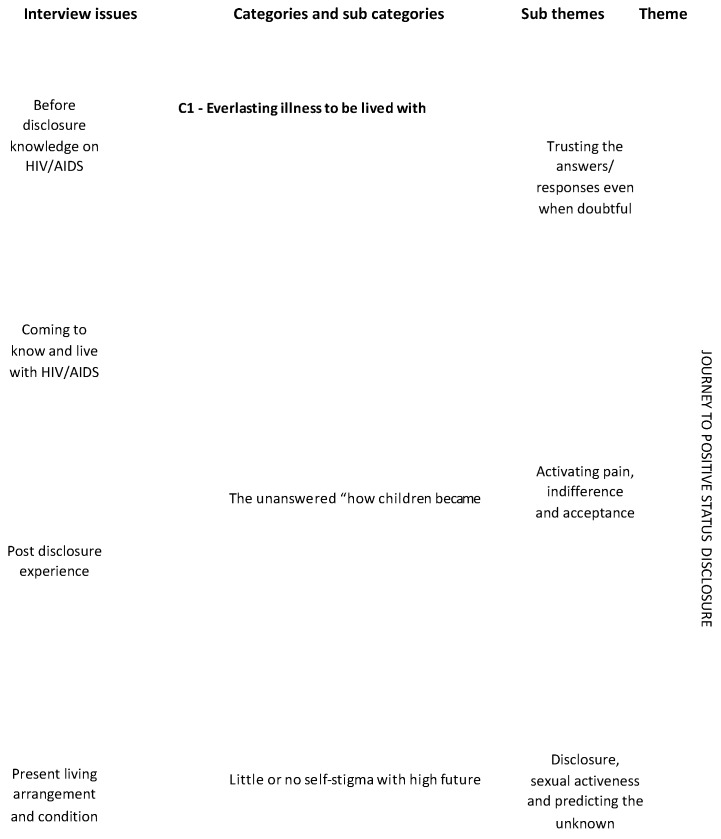
Child/adolescent disclosure pathway context.

**Table 1 ijerph-16-00595-t001:** Child age, geographical setting and region.

No	Region	Urban/Rural	Age (yrs)	Village/Town	ART Site Name	Northern/Southern Ghana
1	Upper East	Rural	11	Fiya	Tongo/	Northern Ghana
2	Upper East	Urban	11	Bolga town	Bolgatanga RH	Northern Ghana
3.	Upper East	Urban	12	Navrongo	Navrongo WMH	Northern Ghana
4.	Upper East	Urban	13	Zuale	Bolgatanga RH	Northern Ghana
5	Upper East	Urban	13	Gambiago	Bolgatanga RH	Northern Ghana
6	Upper East	Rural	14	Sawla	Bongo	Northern Ghana
7	Upper East	Rural	14	Namong Awale	Bongo	Northern Ghana
8	Upper East	Urban	16	Fiya	Tongo	Northern Ghana
9	Upper East	Rural	18	Gamborige	Bongo	Northern Ghana
10	Northern	Rural	10	Savelugu	Walewale	Northern Ghana
11	Northern	Urban	11	Kanvin	Tamale TTH	Northern Ghana
12	Northern	Urban	11	Kanvin	Tamale TTH	Northern Ghana
13	Northern	Rural	11	Walewale	Walewale	Northern Ghana
14	Northern	Urban	12	Central	Tml Central	Northern Ghana
15	Northern	Rural	13	Savelugu	Savelugu	Northern Ghana
16	Northern	Urban	13	Central	Tml Central	Northern Ghana
17	Ashanti	Rural	9	Kyekyebiase	Agogo	Southern Ghana
18	Ashanti	Urban	12	Suntreso	Suntreso	Southern Ghana
19	Ashanti	Urban	13	Bantama	Suntreso	Southern Ghana
20	Ashanti	Rural	14	Agogo free Town	Agogo	Southern Ghana
21	Ashanti	Rural	14	Asaante Akyem Adomfe	Agogo	Southern Ghana
22	Ashanti	Urban	15	Atimatim	Tafo	Southern Ghana
23	Ashanti	Urban	15	Breman Nkontwoma	Tafo	Southern Ghana
24	Ashanti	Urban	16	Pankrono	Tafo	Southern Ghana
25	Ashanti	Rural	16	Asante Akyem	Agogo	Southern Ghana
26	Ashanti	Urban	17	Tafo Medoma	Tafo	Southern Ghana
27	Ashanti	Rural	17	Konongo Odumase	Agogo	Southern Ghana
28	Ashanti	Urban	17	Fawoade	Tafo	Southern Ghana
29	Ashanti	Rural	18	Offinso	Offinso	Southern Ghana
30	Ashanti	Rural	18	Konongo Zongo	Agogo	Southern Ghana

**Table 2 ijerph-16-00595-t002:** Emerged themes and related quotes.

Theme	Sub Theme	Quotes
Pre-disclosure knowledge on HIV and AIDS	Knowing how children became infected with HIV and AIDS	*-Male, 13 years from Gambiago:” I don’t know anything about it”* *-Female, 16 years from Agogo: “…HIV/AIDS is a disease which manifests in the blood; and once there is a blood contact between an infected and non-infected person, the non-infected person will be infected with the virus”* *- Male, 16 years from Tafo:” HIV/AIDS is a disease which manifests in the blood; and once there is a blood contact between an infected and non-infected person, the non-infected person will be infected with the virus”* *-Female, 13 years from Bolgatanga:” I have heard about it but no one has explained to me in details”* *-Female, 15 years from Nkwontwoma:” …yea, there is difference, HIV/AIDS… HIV is not very dangerous but AIDS is very dangerous…Yes, HIV is a virus…Yes, I know my status, ok, I will say HIV is a…. it looks like someone who is having a malaria parasite and went for medicine to reduce the level of malaria…but the AIDS when you get it, it’s the disease”*
The disclosure approach and process		*-Male, 18 years from Tamale:” I had ‘kooko’ and came to the hospital. Upon diagnosis by the Doctor, he disclosed to me about my HIV status. He asked me whether I am aware and I said; no. I have not been informed by anyone…When I was young, I didn’t know the reason for coming for the drugs always. Later I read about the drugs and realized that I have HIV. Again, my mother confirmed to it and some of the nurses later disclosed to me. I got ill and was admitted so was later told”* *-Male, 18 years from Offinso: “When I got ill and was admitted then my parents were also diagnosed”* *- Male, 13 years from Bolgatanga: “I was informed by a nurse”* *-Female 14 years from Agogo: “My grandmother disclosed to me at home. I was not happy about it at all. I came to seek medical attention and never expected to be tested for HIV/AIDS. So, when I was told about it, I could not believe it. It was later that I came for a lab test and it was confirmed”*
Earliness or lateness of disclosure nested in preferred ideal age of disclosure		*-Male, 18 years from Tamale: “When I was 17 that they informed me, …yes, what they did was paining me, getting old and telling me about it, …oh am okay with that”* *-Female, 16 years from Agogo: “They waited for me to take the drugs for some time before disclosing it to me …I was pleased with the process of disclosure”* *-Female, 16 years from Tongo: “…Three years… I was informed too early”* *-Male, 17 years from Sangnerirukuku:” …months, my uncle sent me to the hospital … felt that the doctor has prescribed him the medicines so that’s all. all I know that it has been given to me by doctors, so I think that’s the right thing to do so they know what they are doing …Anytime I ask they tell me those things they always tell me, that I have something in my stomach”*
Reaction and response to disclosure news		*- Female, 16 years from Agogo: “I felt sick and when I was sent to the hospital, it came out that I am HIV/AIDs positive…The doctor first informed my mother and she cried. When I asked her why? she failed to respond to me. Then a nurse called me and asked me whether I have had an intimacy with someone before and I said ‘No’. Then she asked my mother whether she has really found out from me concerning having intercourse with someone and she replied ‘Yes’. Later, the nurse called me and informed me privately. She told me not to feel sad…I felt sad and I still think about it… My mother should have informed me…I tried several times, but she failed to disclose to me”*
Medication intake and adherence	Coping strategy and food insecurity	*-Male, 15 years from Tafo: “I take every day. Morning and evening…I feel healthy when I take the drugs… sometimes when I travel. I remember I went to Accra and stayed for about 8 months of which I did not take the drugs along”* *-Female, 16 years from Agogo:” I take every day---in the mornings and in the evenings…I know that it will strengthen my immune system and the viruses will be destroyed…Yes. Sometimes, I feel I am taking the drugs too frequently so I pulse in some cases. Again, taking the drug is not a pleasant experience, hence I sometimes take in the morning and skip the evening and vice versa…I feel comfortable taking drugs at home. At the work place, I take the drugs secretly so that people will not know”*

**Table 3 ijerph-16-00595-t003:** Comparing HIV and AIDS knowledge levels of study participants.

Interview Issue	Southern Ghana	Northern Ghana
*What do you know about the HIV/AIDs disease?*	*Child 1: “HIV/AIDs is a disease which manifests in the blood; and once there is a blood contact between an infected and non-infected person, the non-infected person will be infected with the virus”* *Child 3: “HIV/AIDs is a disease which affects the immune system”* *Child 4: “It is a disease that is very disturbing”* *Child 5: “HIV/AIDs is a disease which manifests in the blood; and once there is a blood contact between an infected and non-infected person, the non-infected person will be infected with the virus”* *Child 9: “I have heard about it before…I know difference…The difference is that if the HIV comes on you, you can have very quick treatment for it but the AIDS can easily kill you early”* *Child 10: “yea, there is difference, HIV/AIDS, …HIV is not very dangerous, but AIDS is very dangerous”*	*Child 6: “I have heard HIV lives in the body”* *Child 7: “I can’t tell”* *Child 8: “…they say it is a deadly disease but for me I don’t see it as a deadly disease”* *Child 11: “I know it’s a disease”* *Child 13: “I have heard about it but no one has explained to me in details”* *Child 14: “I don’t know anything about it”*

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
