# Peer review of "A Phenomenological Account of HIV Disclosure Experiences of Children and Adolescents from Northern and Southern Ghana"

_ijerph, 2019, doi:10.3390/ijerph16040595_

Round 1
Reviewer 1 Report
In the paper entitled A phenomenological account of HIV disclosure experience of children and adolescents from Northern and Southern Ghana. The authors analyse the children’s HIV disclosure experiences across Northern and Southern Ghana highlighting that the children’s dissatisfaction with the timing of disclosure. This topic is important for both academics and practitioners. The manuscript is divided into the following sections: Introduction, Materials and Method, Results, Discussion, Conclusion.
The paper is interesting and includes noteworthy information to justify publication. However, I would suggest some changes.
Introduction reveals the topic and why it is relevant, but it does not inform about the research method. Furthermore, the study results are not presented, nor is the information on the new knowledge that the paper contributes.
In the sectionMaterials and Method, after first figure, there is a table without description. In this table, in first line, letter “U” is missing. Then you use a content analysis, you follow 6 step but I would suggest explaining more information about these six steps are built.
In the section Results, in table 1, there is an error in the numbers (5 is missing).
In line 412, “When u take the drugs”. What is “u”?
In line 433, “I eat twice a da”. What is “da”?
For your conclusion, you can consider also the following study:
Kouyoumdjian, F. G., Meyers, T., & Mtshizana, S. (2005). Barriers to disclosure to children with HIV. Journal of tropical pediatrics, 51(5), 285-287.
Vreeman, R. C., Gramelspacher, A. M., Gisore, P. O., Scanlon, M. L., & Nyandiko, W. M. (2013). Disclosure of HIV status to children in resource‐limited settings: a systematic review. Journal of the International AIDS Society, 16(1), 18466.
Author Response
Reviewer 1 Comment | Response to reviewer comment |
Introduction reveals the topic and why it is relevant, but it does not inform about the research method. Furthermore, the study results are not presented, nor is the information on the new knowledge that the paper contributes.
| It is my understanding that this comment applies to the abstract. If so, then this paper was written strictly according to the journal style and specification which does not call for sectioning the abstract into introduction, methods, results, and conclusion.
However all these are all embedded in the current format of abstract that was presented.
If it were to be sectioned it will appear like this;
Introduction: Disclosure of HIV status to infected children, though challenged by caregiver dilemma, remains central in achieving the UNAIDS global goal of 90/90/90. This study explores children’s HIV disclosure experiences across Northern and Southern Ghana. Methods: A qualitative interpretative phenomenological design facilitated the recruitment of 30 HIV positive disclosed children and adolescents aged 9-19 years in 12 antiretroviral treatment (ART) centres in Northern and Southern Ghana between January 2017 and June 2018. Data was collected via in-depth interviews. We used phenomenological analysis applying concepts and categories identification, patterns and interconnections searching, mapping, theme building and constant comparative technique to draw conclusions. Disclosure of HIV status to children occurred with little or no preparation. Results: Caregivers intentionally or out of dilemma often prolonged or postponed disclosure to when children aged older. Illness severity and disease progression principally defined the need for disclosure. Children preference for early status disclosure averaged at age 10 was demonstrated despite the initial disclosure experience of shock and disappointment. There was improved medication adherence despite the challenge of limited knowledge about HIV transmission, financial difficulty and food insecurity. Conclusion: Context and culturally adapted pre- and post- disclosure guideline laced with social protection package is needed to support HIV positive children.
This journal does not recommend the above format for the abstract
However, if the comment relates to the background of the study, then the research method is contained under the section
Material and methods from line 104 to line 203.
Information on the new knowledge that the paper contributes to has been detailed in the discussion section. However this comment has been included in line 103 to line 105
‘The findings of this study contributes both to theory and practice of post disclosure experience of children moving beyond caregiver account to lived experiential account by infected children ‘.
|
Is the research design appropriate? Can be improved | The research design has been improved with a comment insertion in line 130 to line 132 |
In the section Materials and Method, after first figure, there is a table without description. In this table, in first line, letter “U” is missing.
when you use a content analysis, you follow 6 step but I would suggest explaining more information about these six steps are built.
| The Table was part of the Fig 1 which explains the map. It has been fixed at the appropriate place and further explanation has been provided in line 114 to line 115
Further explanation has been provided and inserted in line 182 to line 188.
‘The raw data was read more than once and in some cases several times by two researchers initially. Agreed upon codes were used to delineate common observations. The common patterns were identified independently by the two researchers. The emerging themes were then searched for by each researcher after which they were all grouped. In building the thematic data, four of the researchers were engaged to examine the theme data built. The conclusion drawing involved a constant revisit to the field notes, identified and matched patterns, replay of the audio recordings and constant feedback from the researchers’.
|
In the section Results, in table 1, there is an error in the numbers (5 is missing).
| This has been corrected in line 221 |
In line 412, “When u take the drugs”. What is “u”?
| Correction has been made in line 418. U changed to ‘you’ |
In line 433, “I eat twice a da”. What is “da”?
| correction made as reflected in line 439 of revised document |
For your conclusion, you can consider also the following study: Kouyoumdjian, F. G., Meyers, T., & Mtshizana, S. (2005). Barriers to disclosure to children with HIV. Journal of tropical pediatrics, 51(5), 285-287. Vreeman, R. C., Gramelspacher, A. M., Gisore, P. O., Scanlon, M. L., & Nyandiko, W. M. (2013). Disclosure of HIV status to children in resource‐limited settings: a systematic review. Journal of the International AIDS Society, 16(1), 18466.
| Thanks for the works refereed to be considered relative to our conclusion
These studies have been read particularly Vreeman’s systematic review which reports of Kallem’s study of 71 child/dyad caregivers The setting of that study was on single facility in Accra, Korlebu hospital in the capital of Ghana and with a quantitative focus.
Our study is purely qualitative with phenomenological lenses from both southern Ghana of the Ashanti region and the northern Ghana regions of upper east and northern region where there is no empirical reported studies on post disclosure experience of children and adolescent.
We are thus justified in our conclusion of being the first study to examine children experiential account post disclosure and with the findings made. |
Reviewer 2 Report
The paper provides empirical evidence on a matter where such information is relatively limited. It does not as such stand out from other studies, but id adds with additional and valuable empirical evidenceThe study is professionally performed. It relies on an extensive and up to date literature survey, and it applies qualitative methods in a professional way. The findings of the study, based on Ghanese patient's experience, are innovative and contribute to existing knowledge. Although the study and results are not as such very original, the results are sound and concerned with an important disease and thus call for publication.
The study is based on a qualitative survey, which is summarized professionally using standard qualitative methods. Thus, the claims appears novel and convincing (of course conditioned on what can be derived qualitatively; it is my impression that the authors take necessary reservation
Not within the framework and the problem statement set out. This does not preclude that further future research can be recommended; the authors are explicit on that.
A description / discussion of the limitations of a qualitative survey ad its interpretation might always apply. However, I consider this to be a standard thing that should be deferred to standard textbooks and not take up space in journal articles, unless it is expected that the audience of the journal is less familiar with the method. If you consider such a discussion valuable for your audience, you may ask the authors to add it.
Author Response
Reviewer 2 comment | |
A description / discussion of the limitations of a qualitative survey ad its interpretation might always apply. However, I consider this to be a standard thing that should be deferred to standard textbooks and not take up space in journal articles, unless it is expected that the audience of the journal is less familiar with the method. If you consider such a discussion valuable for your audience, you may ask the authors to add it | The limitations of the qualitative design has been included in line 553 to line 558 See comment below
‘Consistent with qualitative studies, our findings are limited by being context specific for which interpretation may have to be done with some carefulness. Though several attempts were made to ensure accurate and pure bracketing-an essential requirement essential when using the phenomenological design study, the adoption of research translators, different study sites selection with unique cultural context, may not have been able to ensure a full compliance pure bracketing’.
|
English language and style are fine/minor spell check required can be improved
| There were minor spelling errors. These have been addressed. Where addressed, it has been indicated using track changes |